# Amodal Instance Segmentation with Diffusion Shape Prior Estimation

Minh Tran[1], Khoa Vo[1], Vuong Ho[1], Tri Nguyen[2], Ngan Le[1]

[1]AICV Lab, University of Arkansas

[2]Cruise, LLC

## Abstract

*Amodal Instance Segmentation (AIS) presents an intriguing challenge, including the segmentation prediction of both visible and occluded parts of objects within images. Previous methods have often relied on shape prior information gleaned from training data to enhance amodal segmentation. However, these approaches are susceptible to overfitting and disregard object category details. Recent advancements highlight the potential of conditioned diffusion models, pretrained on extensive datasets, to generate images from latent space. Drawing inspiration from this, we propose AISDiff with a Diffusion Shape Prior Estimation (DiffSP) module. AISDiff begins with the prediction of the visible segmentation mask and object category, alongside occlusion-aware processing through the prediction of occluding masks. Subsequently, these elements are inputted into our DiffSP module to infer the shape prior of the object. DiffSP utilizes conditioned diffusion models pretrained on extensive datasets to extract rich visual features for shape prior estimation. Additionally, we introduce the Shape Prior Amodal Predictor, which utilizes attention-based feature maps from the shape prior to refine amodal segmentation. Experiments across various AIS benchmarks demonstrate the effectiveness of our AISDiff.*

## 1. Introduction

Amodal perception, as described in [10], describe human's remarkable ability to perceive objects in their entirety despite occlusion. Building upon this concept, the pioneering studies by [12, 30] introduced amodal instance segmentation (AIS). This approach aims to predict the complete shape of objects, encompassing both their visible and occluded regions. Indeed, AIS exhibits vast potential across various domains, as evidenced by its applications in robot manipulation [1] and autonomous driving [19]. Across various AIS benchmarks [3, 19, 30], a multitude of approaches addressing the AIS challenge have emerged in the literature. These approaches, as evidenced by numerous studies [3, 7, 12, 14, 19, 24, 25], demonstrate the ongoing efforts to

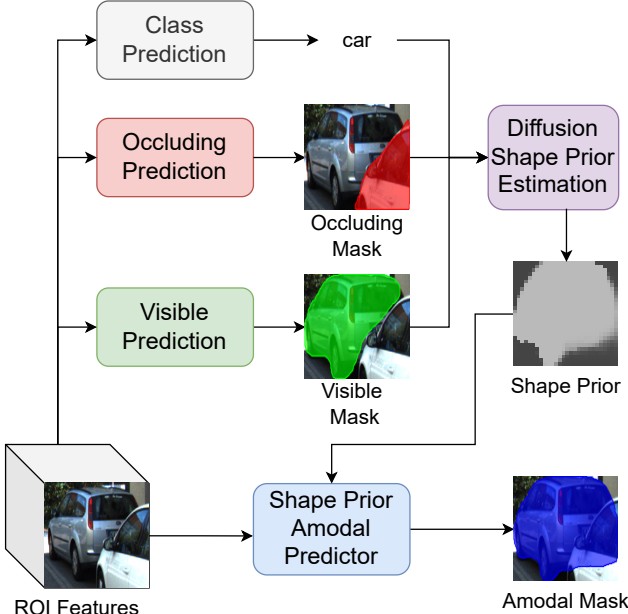

Figure 1. Overall architecture of AISDiff. AISDiff predicts the visible segmentation mask and the object category while simultaneously addressing occlusion by predicting the occluding mask. Next, these predictions are integrated into the Diffusion Shape Prior Estimation (DiffSP) module to establish the object's shape prior. This shape prior is then utilized by AISDiff to produce the amodal segmentation.

tackle this challenge.

Recent research [2, 4, 7, 26, 27] highlights the effectiveness of integrating shape prior information in AIS. Indeed, These shape prior AIS methods typically construct shape-prior knowledge from the training dataset, which is later utilized to train the AIS model. In [26], for instance, the authors employ variational autoencoders to reconstruct amodal masks. The concept revolves around using ground truth amodal masks, utilizing autoencoders to reconstruct them, and storing the encoded codebooks as shape priors. Similarly, in [4], the authors also construct a shape prior codebook but employ a vector-quantization variational au-

toencoder. After establishing the shape prior, these method first predict the coarse amodal segmentation and refine the final amodal segmentation mask using the built shape prior. However, there are limitations to these approaches. Firstly, the shape prior tends to overfit to the training data, consequently leading to overfitting in amodal mask prediction overall. Secondly, since the shape prior is built solely from ground truth amodal masks, it may overlook the object category, which could provide significant supplementary information for deriving the shape prior.

To tackle these issues, we desgin a **AIS** mask head with **Diff**usion Shape Prior Estimation (**AISDiff**). The design of AISDiff is depicted in Figure 1. In essence, AISDiff begins by predicting the visible segmentation mask and the category of the object of interest. Simultaneously, it conducts occlusion-aware processing by predicting the occluding mask, which is the segmentation of occluding elements within the specified ROI. Subsequently, these three pieces of information are fed into the proposed Diffusion Shape Prior Estimation (DiffSP) module to derive the shape prior of the object. Finally, leveraging this shape prior, AISDiff generates the amodal segmentation.

Specifically, DiffSP leverages the successes of conditioned diffusion models (such as Stable Diffusion [20] and GLIDE [17]), which are pretrained on extensive language vision datasets like LAION [22]. This enables the model to capture rich visual features, making it suitable as prior knowledge for downstream tasks [18, 28]. Building upon this foundation, we feed a trained conditioned diffusion model with an ROI image containing only the visible pixels of the object of interest, expecting the model to generate the missing parts. Additionally, an occluding mask and a textual description of the object category is also feed to condition the mdoel. Subsequently, the denoising process iterates $T$ steps to output the generated image containing the occluded parts. However, rather than relying on the final generated pixels, DiffSP exploits on the attention mechanism between the conditioning information and the image features. This attention map remains relatively stable across time steps, thereby reducing the denoising time needed to obtain the shape prior. Furthermore, we design the Shape Prior Amodal Predictor, which learns the attention-based amodal feature map from the acquired shape prior to predict the amodal mask segmentation.

In summary, our contributions are as follows:

- We present AISDiff, a novel AIS mask head featuring a Diffusion Shape Prior Estimation module. This model predicts the visible segmentation mask and category of the object while considering occlusion. It then uses these predictions to estimate the shape prior of the object before generating the final amodal segmentation mask.
- We propose DiffSP module, harnessing the efficacy of conditioned diffusion models to derive the shape prior of the object of interest.
- We introduce the Shape Prior Amodal Predictor, which learns attention-based amodal feature maps from the obtained shape prior to predict the amodal segmentation.

## 2. Related Work

Amodal instance segmentation involves predicting an object's shape, including both its visible and occluded parts. Li and Malik [12] pioneered a method aimed at addressing AIS. They proposed enlarging the modal bounding box in alignment with high heatmap values and synthesizing occlusions. Following this seminal work, various methodologies have surfaced in literature. Notably, ORCNN [3] introduces instance mask heads for both amodal and visible instances, along with an additional head for predicting occluded masks. ASN [19] builds upon ORCNN by integrating a multi-level coding module for bidirectional feature modeling of visible and amodal aspects. BCNet [8] enhances amodal mask prediction by incorporating a supplementary branch dedicated to predicting occlusion masks within the bounding box. AISFormer [24] introduces a transformer-based mask head, demonstrating the efficacy of transformer modeling in generating AIS masks. However, their approach, which consolidates all mask relationships into one transformer model, leads to compromised visible segmentation output, consequently affecting the quality of amodal segmentation output due to bidirectional feature relations as mentioned earlier.

Recent studies [7, 26] underscore the benefits of integrating shape priors into AIS. These methods leverage prior knowledge of mask shapes to improve amodal mask predictions. VRSP-Net [26] predicts coarse amodal masks, retrieves shape priors using a simple autoencoder, and then refines the final amodal mask predictions. AmodalBlastomere [7] employs a similar strategy with a variational autoencoder for blastomere and cell segmentation. Despite their progress, these methods often overlook the importance of object categories when utilizing prior shapes. Moreover, their training procedures frequently lead to overfitting of the shape prior model to the training dataset. Additionally, these approaches simply incorporate the shape prior by concatenating it with visible features to refine amodal masks.

## 3. Method

### 3.1. Overall AIS Setup

Given an input image $\mathbf{I}$, we follow most of previous AIS settings [3, 8, 24, 26], utilizing a pre-trained backbone network, such as ResNet [6], RegNet [21] to extract spatial visual representation. An object detector such as FCOS [23], or Faster-RCNN[6], can be subsequently adopted to obtain $n$ regions of interest (RoI) predictions and their corresponding visual features $\{\mathbf{F}^i\}_{i=1}^n$. Follow most of previous works

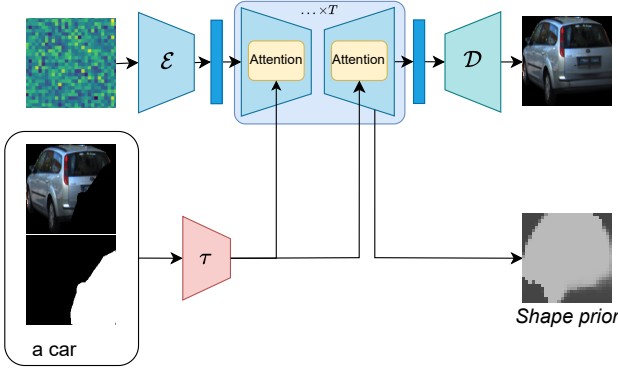

Figure 2. Overall process of Diffusion Shape Prior Estimation (DiffSP).

[8, 24, 26], the object detector being chose is Faster R-CNN for fair comparison. Here, each RoI is presented by its visual feature $\mathbf{F}^i \in \mathbb{R}^{C_e \times H_r \times W_r}$, where $C_e$ denotes the feature channel size and $H_r \times W_r$ represents the spatial shape of the pooling feature. In this context, given a RoI, AISDiff takes $\mathbf{F}^i$ as input and aims to predict the amodal mask $\mathbf{M}_a^i$. Moreover, in this case, we also denote the visible mask $\mathbf{M}_v^i$, and the occluding mask $\mathbf{M}_o^i$.

## 3.2. AISDiff

The overall design of AISDiff is depicted in Fig. 1. Initially, we discuss the prediction process for the visible segmentation of the object of interest, along with its categories, incorporating occlusion-awareness through the prediction of occluding masks (Sec. 3.2.1). Following this, we introduce the DiffSP method in detail (Sec. 3.2.2). Lastly, we present the Shape Prior Amodal Predictor (Sec. 3.2.3).

### 3.2.1 Occlusion-aware Visible Segmentation

Given the ROI feature $\mathbf{F}^i$, AISDiff first aims to predict the visible segmentation mask and the category of the object of interest, while simultaneously conducts occlusion-aware ability by predicting the occluding mask, which is the segmentation of occluding elements within the specified ROI. BCNet [8] is utilized as the foundation for the Occlusion-aware Visible Segmentation module. This module consists of two branches: one for occluding mask prediction and the other for visible mask prediction. Drawing from the methodology outlined in [8], both branches follow a similar design structure, encompassing two main components: feature extraction and mask prediction. The feature extraction segment comprises a sequence of layers, including a $3 \times 3$ convolutional layer with a stride of 1, a Graph Convolutional Network (GCN) [11] block, and another $3 \times 3$ convolutional layer with a stride of 1. Subsequently, the mask prediction component is constructed with a $2 \times 2$ transposed

convolutional layer employing a stride of 2, coupled with a $1 \times 1$ convolutional layer using a stride of 1.

Furthermore, to enhance occlusion awareness and subsequently improve visible segmentation accuracy, features extracted from the occluding branch are incorporated into the ROI feature $\mathbf{F}^i$ before being fed into the feature extraction section of the visible branch. Simultaneously, features extracted from the visible branch are utilized for object category prediction. This classification step employs a fully connected layer with an output dimension corresponding to the number of categories present in the datasets under consideration. In summary, the final output of this module comprises the visible mask $\mathbf{M}_a^i$, the occluding mask $\mathbf{M}_o^i$, and the object category $c^i$.

### 3.2.2 DiffSP

The process depicted in Fig. 2 illustrates the Shape Prior Estimation (DiffSP) module. DiffSP builds upon the successes of conditioned diffusion models, such as Stable Diffusion [20] and GLIDE [17], which are pre-trained on comprehensive language-vision datasets like LAION [22]. This pre-training equips the model with the ability to capture intricate visual features, rendering it suitable as prior knowledge for subsequent tasks [18, 28]. Expanding on this foundation, DiffSP utilizes a trained conditioned diffusion model and inputs a ROI image containing only the visible pixels of the object under consideration, expecting the model to generate the obscured parts. Additionally, the model is conditioned with an occluding mask and a textual description of the object category. Subsequently, the denoising process iterates $T$ steps to produce the generated image containing the occluded regions. However, instead of relying solely on the final generated pixels, DiffSP capitalizes on the attention mechanism between the conditioning information and the image features.

Specifically, Stable Diffusion [20] is employed as the pre-trained conditioned diffusion model, leveraging its self and cross-attention layers. Specifically, the random Gaussian noise is encoded into latent space and then experiences the denoising process over $T$ time steps to generate the inpainting image. In fact, the ROI image containing only the visible pixels of the object of interest, the occluding mask, and the textual description of the object category serve as conditions and are represented as $y$, which is projected by $\tau$ into an intermediate representation $\tau(y)$. At each denoising step $t$, a UNet architecture with $L$ layers of self and cross-attention transforms $z_t$ into $z_{t-1}$. Specifically, at layer $l$ and time step $t$, the cross-attention layer captures the relationship between $z_t$ and the encoded condition $\tau(y)$, reflecting the entire reconstructed shape of the object. This relationship is formalized as follows: at layer $l$ and time step $t$, the self-attention map is denoted as $\mathcal{A}_S^{l,t}$, and the cross-

attention map is denoted as $\mathcal{A}_C^{l,t}$. Moreover, as demonstrated in [16], the attention map remains relatively stable across time steps. Following the methodology of [16], we average these cross and self-attention maps over layers and time steps, setting $T = 10$. Additionally, as also suggested in [16], although the cross-attention maps $\mathcal{A}_C$ already outline the shape of the reconstructed object, they tend to be coarse-grained and noisy. To refine the precision of object localization, we follow [16], utilizing the self-attention map $\mathcal{A}_S$ to enhance $\mathcal{A}_C$. Consequently, the shape prior is obtained by: $\mathbf{M}_{sp} = (\mathcal{A}_S)^\tau \cdot \mathcal{A}_C$.

### 3.2.3 Shape Prior Amodal Predictor

The design of Shape Prior Amodal Predictor is depicted in Fig. 3. Initially, the feature extraction module utilizes the ROI feature $\mathbf{F}^i$ to generate the amodal feature. This module is constructed using a sequence of $3 \times 3$ convolutional layers with a stride of 1. Subsequently, the obtained amodal feature undergoes processing in the attention learning module in conjunction with the shape prior $\mathbf{M}_{sp}$ obtained from DiffSP, aimed at learning the spatial attention map. Specifically, the attention computation involves passing the amodal feature through a sequence of $3 \times 3$ convolutional layers with a stride of 1, followed by a sigmoid activation function. This computed attention map is then multiplied with the shape prior $\mathbf{M}_{sp}$. The spatial attention map is further multiplied with the amodal feature to obtain the attention amodal feature. This feature is then fed into a mask prediction module, which is structured with a $2 \times 2$ transposed convolutional layer employing a stride of 2, coupled with a $1 \times 1$ convolutional layer using a stride of 1, to derive the amodal mask $\mathbf{M}_a^i$

### 3.3. Objective Function & Training

Employing AIS protocols, the training adopts a two-stage instance segmentation process similar to Mask R-CNN, facilitating concurrent training of both bounding box and amodal mask prediction heads alongside the object detection framework. In essence, the training procedure optimizes a multi-task loss function $\mathcal{L}$ as follows:

$$\mathcal{L} = \mathcal{L}_{det} + \mathcal{L}_{cls} + \mathcal{L}_v + \mathcal{L}_o + \mathcal{L}_a \qquad (1)$$

where $\mathcal{L}_{det}$ is object detection loss, defined similarly to that in Faster R-CNN object detection. The occluding mask loss $\mathcal{L}_o$, the visible mask loss $\mathcal{L}_v$, the amodal mask loss $\mathcal{L}_a$, and the classification loss $\mathcal{L}_{cls}$ are computed using cross entropy loss with the corresponding ground truth.

## 4. Experiments

### 4.1. Datasets, Metrics and Implementation Details

**Datasets:** We benchmark our AISDiff on three AIS datasets, namely KINS [19], COCOA-cls [3], and D2SA

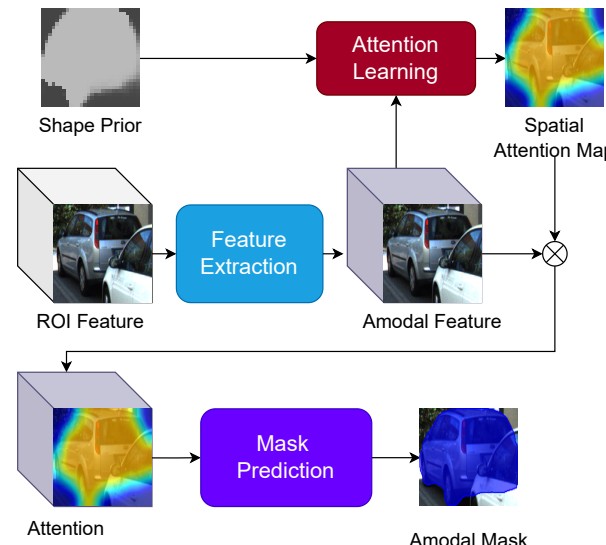

Figure 3. Overall design of Shape Prior Amodal Predictor.

Table 1. *Performance comparison on KINS test set* with various backbones. † indicates our reproduced results.

| Backbones & Methods | Venue | Shape Prior | $AP \uparrow$ | $AP_{50} \uparrow$ | $AP_{75} \uparrow$ | $AR \uparrow$ |
|---|---|---|---|---|---|---|
| **ResNet-50** | | | | | | |
| PCNet[29] | CVPR20 | ✗ | 29.1 | 51.8 | 29.6 | 18.3 |
| ASBU[15] | ICCV21 | ✗ | 29.3 | 52.1 | 29.7 | 18.4 |
| Mask R-CNN[9] | ICCV17 | ✗ | 30.0 | 54.5 | 30.1 | 19.4 |
| ORCNN[3] | WACV19 | ✗ | 30.6 | 54.2 | 31.3 | 19.7 |
| ASN[19] | CVPR19 | ✗ | 32.2 | - | - | - |
| AISFormer[24] | BMVC22 | ✗ | 33.8 | 57.8 | 35.3 | 21.1 |
| AmodalBlastomere[7] | TMI20 | ✓ | 30.3 | - | - | - |
| VRSP-Net[26] | AAAI21 | ✓ | 32.1 | 55.4 | 33.3 | 20.9 |
| **AISDiff (Ours)** | - | ✓ | **33.9** | **58.8** | **35.2** | **22.0** |
| **ResNet-101** | | | | | | |
| Mask R-CNN[6] † | ICCV17 | ✗ | 30.2 | 54.3 | 30.4 | 19.5 |
| BCNet[8] | CVPR21 | ✗ | 28.9 | - | - | - |
| BCNet[8] † | CVPR21 | ✗ | 32.6 | 57.2 | 35.4 | 21.5 |
| AISFormer[24] | BMVC22 | ✗ | 34.6 | 58.2 | 36.7 | 21.9 |
| **AISDiff (Ours)** | - | ✓ | **35.1** | **58.8** | **37.3** | **23.0** |
| **RegNet** | | | | | | |
| ASPNet[14] | CVPR22 | ✗ | 35.6 | - | - | - |
| AISFormer[24] | BMVC22 | ✗ | 35.6 | 59.9 | 37.0 | 22.5 |
| **AISDiff (Ours)** | - | ✓ | **36.1** | **60.1** | **38.6** | **23.0** |

[3]. KINS is a large-scale traffic dataset with 95,311 training instances and 92,492 testing instances with 7 categories. COCOA-cls is an AIS dataset that is derived from MSCOCO [13] with 80 categories of 6,763 training instances and 3,799 testing instances. D2SA is an AIS dataset with 60 categories of instances related to supermarket items with 13,066 training instances and 15,654 testing instances.

**Metrics:** Following existing AIS methods [24, 26], we adopt mean average precision (AP) and mean average recall (AR).

Table 2. *Performance comparison on D2SA test set* with ResNet-50 as backbone. † indicates our reproduced results.

| Methods | Venue | Shape Prior | AP ↑ | $AP_{50}$ ↑ | $AP_{75}$ ↑ | AR ↑ |
|---|---|---|---|---|---|---|
| Mask R-CNN[6] | ICCV17 | ✗ | 63.57 | 83.85 | 68.02 | 65.18 |
| ORCNN[3] | WACV19 | ✗ | 64.22 | 83.55 | 69.12 | 65.25 |
| ASN[19] † | CVPR19 | ✗ | 63.94 | 84.35 | 69.57 | 65.20 |
| BCNet[8] † | CVPR21 | ✗ | 65.97 | 84.23 | 72.74 | 66.90 |
| AISFormer[24] | BMVC22 | ✗ | 67.22 | 84.05 | 72.87 | 68.13 |
| VRSP-Net[26] | AAAI21 | ✓ | 70.27 | 85.11 | 75.81 | 69.17 |
| **AISDiff (Ours)** | - | ✓ | **71.01** | **85.12** | **76.23** | **69.29** |

Table 3. *Performance comparison on COCOA-cls test set*, ResNet-50 as backbone. † indicates our reproduced results.

| Methods | Venue | Shape Prior | AP ↑ | $AP_{50}$ ↑ | $AP_{75}$ ↑ | AR ↑ |
|---|---|---|---|---|---|---|
| Mask R-CNN[6] | ICCV17 | ✗ | 33.67 | 56.50 | 35.78 | 34.18 |
| ORCNN[3] | WACV19 | ✗ | 28.03 | 53.68 | 25.36 | 29.83 |
| ASN[19] † | CVPR19 | ✗ | 35.33 | 58.82 | 37.10 | 35.50 |
| BCNet[8] † | CVPR21 | ✗ | 35.14 | 58.84 | 36.65 | 35.80 |
| AISFormer[24] | BMVC22 | ✗ | 35.77 | 57.95 | 38.23 | 36.71 |
| VRSP-Net[26] | AAAI21 | ✓ | 35.41 | 56.03 | 38.67 | 37.11 |
| **AISDiff (Ours)** | - | ✓ | **35.93** | **58.86** | **38.63** | **37.14** |

## 4.2. Performance Comparison

### 4.2.1 Quantitative Results

**KINS.** Tab. 1 depicts the comparison between AISDiff and SOTA AIS methods on the KINS dataset. AISDiff demonstrates consistent improvements across various backbones, including ResNet [5], ResNet [5], and RegNet [21]. Specifically, when compared to methods utilizing ResNet-50 as the backbone, our method outperforms both SOTA methods that use shape prior (e.g., and VRSP-Net [26] by 1.8 AP) and methods that do not use shape prior (e.g., AISFormer [24] by 0.1 AP), respectively. When ResNet-101 is utilized as the backbone, our method achieves a improvement over AISFormer, outperforming it by 0.5 AP. Furthermore, compared to APSNet [14] and AISFormer [24] on the RegNet backbone, our approach achieves SOTA performance by surpassing them 0.5 AP.

**D2SA.** Tab. 2 further validates our approach on D2SA dataset. We achieve best results across all metrics. Specifically, we gains 0.74 on AP and 0.12 AR in comparison with the second best method, i.e. VRSP-Net.

**COCOA-cls.** Tab. 3 shows our results on COCOA-cls dataset. Our AISDiff also outperform other methods on all metrics. In fact, it outperforms the second best by 0.16 AP and 0.03 AR.

### 4.2.2 Qualitative Results

Fig. 4 illustrates the qualitative output of AISDiff. The results are arranged from left to right, encompassing: in-

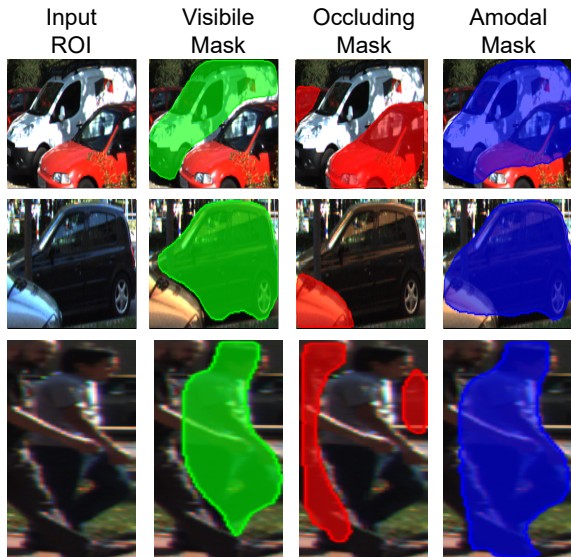

Figure 4. Qualitative results of AISDiff. Left to right: Input RoI, Visible masks, Occluding masks, Amodal masks. Best viewed in color.

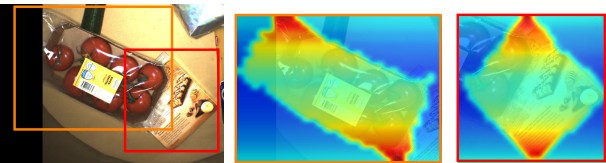

Figure 5. Spatial attention map of the Shape Prior Amodal Predictor on the each RoI. Best viewed in color.

put ROIs, Visible Masks, Occluding Masks, and Amodal Masks.

Fig. 5 visualizes the spatial attention map of the Shape Prior Amodal Predictor on ROIs of the image. The attention maps are well-constrained to the object shape. Moreover, we can see that the decoder typically attends to the visible parts of objects that are similar to the occluded regions when predicting the amodal mask.

## 5. Conclusion

In conclusion, we propose AISDiff, an AIS mask head with a Diffusion Shape Prior Estimation module. This module, termed DiffSP, leverages pre-trained conditioned diffusion models on extensive datasets to extract nuanced visual features for deriving the shape prior of the object. Furthermore, we present the Shape Prior Amodal Predictor, which utilizes attention-based feature maps from the shape prior to enhance amodal segmentation. Through extensive experimentation across diverse AIS benchmarks, we affirm the efficacy of AISDiff.

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
