# OpenReview forum: "Amodal Instance Segmentation with Diffusion Shape Prior Estimation"
_thecvf.com/CVPR/2024/Workshop/POETS — CVPR 2024 Workshop POETS Poster_

### Official Review · Reviewer_hdKM · 2024-04-30

**Rating:** 7
**Confidence:** 4

**Review:**

Summary:

This paper introduces Diffusion-based methods into Amodal Instance segmentation for the first time. By leveraging the shape knowledge, the authors propose Diffusion Shape Prior Estimation (DiffSP) to estimate the potential shape knowledge of various object for amodal mask prediction. Moreover, the proposed method can be applied on various detectors and show the effectiveness on three different datasets.


Strength:

1, The first work that explores the diffusion model for amodal segmentation and mask prediction.

2, The motivation is clear since diffusion models are pre-trained with huge object level data which can be used to predict missing masks or shape of amodal objects.

3, The overall writing is good and easy to understand.

4, The proposed method achieves consistent improvements on various detectors and datasets, which is convincing to me.


Weakness:

1, Missing analysis on computation cost or run time analysis.

2, Better to show different diffusion model’s effects, such as Stable-XL.

---

### Official Review · Reviewer_nG9B · 2024-05-03

**Rating:** 6
**Confidence:** 4

**Review:**

* Summary

This paper addresses amodal instance segmentation (AIS), which aims to recover wholes from occluded parts in images. AIS methods typically leverage shape priors for segmentation. However, prior works often rely on shape priors learned from a limited number of unlabeled masks, which are prone to overfitting and neglect class information. To address this, the paper proposes using text-to-image (T2I) diffusion models trained on a large-scale dataset with semantics. The proposed method infers class and condition it into the T2I diffusion to obtain shape priors, which are used to generate final AIS masks. This approach outperforms prior works not using diffusion priors.


* Strength

The 3D or shape prior abilities of T2I diffusion models have been reported in many research papers. Therefore, it is reasonable to explore how to properly utilize the information for the AIS problem.


* Weakness

The baseline methods are outdated; all of them are from before 2022. There are a bunch of works that utilize diffusion priors for the AIS problem [1, 2, 3]. The paper should compare with those approaches both methodologically and empirically.

[1] Ozguroglu et al. pix2gestalt: Amodal Segmentation by Synthesizing Wholes. CVPR 2024.\
[2] Zhan et al. Amodal Ground Truth and Completion in the Wild. CVPR 2024.\
[3] Xu et al. Amodal Completion via Progressive Mixed Context Diffusion. CVPR 2024.

---

### Official Review · Reviewer_g2qZ · 2024-05-06
**Review of Amodal Instance Segmentation with Diffusion Shape Prior Estimation**

**Rating:** 7
**Confidence:** 3

**Review:**

** Summary **

This paper proposes a method to address Amodal Instance Segmentation. To mitigate the overfitting issue of relying on shape prior information from previous research, this approach uses a large-scale pretrained diffusion model. By introducing a new architecture with the Diffusion Shape Prior Estimation module, it enhances the performance of amodal instance segmentation.

** Pros **

- (Significance) The paper addresses a significant problem of amodal instance segmentation (AIS) that AIS has various applications.
- (Clarity)  The introduction of prior work and the clear presentation of methods and experiments make it easy to read.
- (Method) The authors effectively leverage the powerful ability of large-scale pretrained models for shape prior estimation, presenting a logically sound approach that is easy to follow.
- (Effectiveness) Performance has been proven across various benchmarks, including KINS, COCOA-cls , and D2SA.

** Cons **

- (Originality) While the novelty is not surprisingly remarkable, the addition of a new module based on existing work has improved performance.
- (Effectiveness) Considering the complex added module DiffSP and Shape Prior Amodal Predictor, the performance improvement is somewhat marginal compared with the methods [14, 24] which do not require shape prior.
- (Minor correction): Line283: four AIS datasets -> three AIS datasets.

---

### Meta-Review · Program_Chairs · 2024-05-14

**Recommendation:** Accept (Poster)
**Confidence:** 5

**Metareview:**

This paper introduces a novel approach using diffusion models for amodal instance segmentation (AIS), showing promising results on multiple benchmarks. It enhances AIS performance by leveraging large-scale pre-trained models for shape estimation. While the method is an improvement over existing techniques, the marginal performance gain and lack of comparison with recent relevant works are noted weaknesses. Also, the relationship between humans and segmentation in city scenes should be discussed to better match this workshop's topic.
Overall, the approach is innovative and the paper is well-written, warranting acceptance with some reservations regarding originality and computational efficiency analysis.

---

### Decision · Program_Chairs · 2024-05-14

Accept (Poster)